# Family caregivers' experience of caring for patients undergoing hemodialysis: A qualitative study at Muhimbili National Hospital in Dar es Salaam, Tanzania

**Ally Halfan Magenge**[1●], **Menti Lastone Ndile**[1●]*, **Francis Furia**[2]

**1** Department of Clinical Nursing, School of Nursing, Muhimbili University of Health and Allied Sciences, Dar es Salaam, Tanzania, **2** Department of Paediatrics and Child Health, School of Clinical Medicine, Muhimbili University of Health and Allied Sciences, Dar es Salaam, Tanzania

● These authors contributed equally to this work.
* menti.ndile@muhas.ac.tz

## Abstract

### Background

The number of patients with chronic kidney disease requiring hemodialysis services is growing in Tanzania. Considering patients undergoing hemodialysis are outpatients, family caregivers fill the gap of caring and support during the lifelong treatment process. Limited information exists especially in the context of Tanzania regarding the challenges the family caregivers encounter during their involvement in the caring process. This study aimed to explore family caregivers' experience of caring for patients undergoing hemodialysis at Muhimbili National Hospital, Dar es Salaam, Tanzania.

### Methods

A qualitative study employing phenomenological design was conducted. Fourteen family caregivers of patients receiving hemodialysis services at Muhimbili National Hospital were purposively selected. Semi-structured in-depth interviews were used for data collection. Interviews were audio-recorded and transcribed verbatim. Data were analyzed following Braun and Clarke's six phases of thematic analysis.

### Results

The study uncovered four key themes from fourteen interviews. The themes were: 1) Family caregivers and providers' interaction, focusing on interpersonal and professional relationships 2) The status of dialysis treatment, describing access and availability of resources for the provision of hemodialysis services 3) Impact of long-term care, touching issues of emotional, physical and social-economic impact to family members and 4) Coping with caring, describing internal and external mechanisms family members use to adjust to demanding situation of caring their loved ones.

**Data availability statement:** All relevant data are within the paper and its Supporting Information files.

**Funding:** The author(s) received no specific funding for this work.

**Competing interests:** The authors have declared that no competing interests exist.

**Abbreviations:** CKD, Chronic Kidney Disease; ESKD, End-Stage Kidney Disease; FCGs, Family Care Givers; IRB, Institutional Review Board; MNH, Muhimbili National Hospital; MUHAS, Muhimbili University of Health and Allied Sciences.

## Conclusion

Caring for patients receiving hemodialysis treatment burdens family caregivers, impacting their livelihood and resources. Healthcare providers need to build a close working relationship with family caregivers to understand their challenges and needs and help build their coping capacities. Additionally, dialysis service coverage issues such as affordability, accessibility, and availability need to be addressed to reduce the caring burden on family caregivers.

## Background

Chronic kidney disease (CKD) is a global public health problem affecting approximately 10% of the global population with a higher burden among people living in low and middle-income countries (LMIC [1,2]. The prevalence of CKD in Sub-Saharan Africa is estimated to be 14% [3] which is similar to the prevalence in Tanzania estimated to be between 7 and 16% [4]. By 2024, CKD is projected to become the fifth leading cause of years of life lost worldwide, with a significant impact on LMICs [5].

Hemodialysis is one of the main treatment options for patients with CKD, however, it has limited access for many patients in LMICs [6]. Many countries in LMIC settings have made efforts to improve access to care, this has been the case for Tanzania which has seen an increase in the number of hemodialysis units and patients on maintenance therapy from about 1000 in 2019 to more than 3000 [7,8]. However, despite country-specific efforts to improve dialysis access, statistics indicate that fewer than 5% of patients with CKD in Sub-Saharan Africa receive life-saving dialysis treatment [1].

Family caregivers (FCGs) play a key role in supporting and caring for patients undergoing hemodialysis by attending to their psychological, social, physical, and emotional needs. The caring responsibilities shouldered by FCGs are reported to hurt their quality of life straining their physical, social economic, and psychological capacities which may consequently lead to poor coping with life [9–12].

Quality of care of patients on maintenance hemodialysis requires a comprehensive approach that will support FCGs. The growing number of patients with CKD on dialysis calls for initiatives for supporting FCGs. While studies in LMICs have highlighted health system challenges faced by FCGs, a significant gap remains in understanding FCGs' experiences within Tanzania's context. Specifically, there is no data on how Tanzania's unique cultural dynamics—such as gender roles, communal caregiving traditions, or reliance on extended family networks shape FCGs' experiences in caring for chronic diseases like CKD. Without studies exploring these contextual factors, initiatives to support FCGs risk being ineffective or culturally misaligned, perpetuating cycles of caregiver burnout and hindering Tanzania's ability to manage its growing CKD burden.

The qualitative study was conducted aiming to explore the experiences of FCGs of patients undergoing hemodialysis treatment at Muhimbili National Hospital. This approach is suited to uncovering relational dynamics and understanding the meaning the FCGs attach to their caring role.

## Methods

### Study design

This was a descriptive qualitative study using semi-structured interviews to explore the FCGs' experiences in caring for their family members undergoing hemodialysis. Reflexive thematic analysis approach was used in analyzing data [13]. The Consolidated Criteria for Reporting

Qualitative Research (COREQ) checklist [14] was employed to enhance the rigor and transparency of the study's reporting, as detailed in Supporting Information S1 Text.

## Study setting

The study was conducted in the renal unit at Muhimbili National Hospital (MNH), a national referral hospital in Tanzania. The unit is the largest and more advanced in the country equipped with 46 hemodialysis machines serving an average of 138 patients per day. Generally, patients at the unit receive 4 hours of hemodialysis sessions thrice a week.

## Sample size and selection

Participants in this study were purposefully selected [15], considering that they are over 18 years old and accompanying their family members to the renal unit. Additionally, to ensure consistency and richness of information provided, only participants with at least six months of caregiving and who could express themselves well were selected. Based on the literature, initially, 20 participants were considered for recruitment [16], however, recruitment of participants ended at 14 participants after reaching saturation point, when no new pattern or insight emerged from additional interviews [17]. This was observed through constant comparison of new data to previously coded data from interviews.

## Data collection and procedure

Data collection took place from 5th June to 1st July 2023. Eligible FCGs escorting their patients to the renal unit for hemodialysis were identified, and were given information about the purpose of this study while waiting for their patients undergoing dialysis sessions; those who agreed to participate were given appointments for interview based on their convenience for their subsequent visit to the renal unit. On the scheduled dates, participants were asked to provide written informed consent to participate in the study, and those who consented were interviewed in a dedicated room in the renal clinic to ensure privacy. One researcher (AM) conducted the interviews using a semi-structured interview guide developed by the research team based on the topic and literature and pretested to two FCGs (See Supporting Information S2 Text). Only the researcher and one participant were in the dedicated interview room. After self introduction and interview process explained, the opening question was *Tell me about your life when you started caring for your loved one in need of hemodialysis treatment*. The conversation with respective FCGs flowed naturally by asking questions followed by probes, no FCG withdrew from the interview. All interviews were audio-recoded and field notes were taken to capture non-verbal responses and other emerging issues. The interviews lasted between 45–70 minutes. Initial interviews were used by researchers to shape the subsequent interviews to ensure the richness of the collected data. All Interviews were conducted in Kiswahili language which is the national language of the participants.

## Data analysis

The study employed an inductive thematic analysis approach following Braun and Clarke's six phases of thematic analysis [18]. The audio-recorded interviews were transcribed verbatim and transcripts were then compared with audio files to ensure no data was missing. A reflexive thematic analysis trail was followed [13], whereby a researcher is actively involved in interpretations of patterns of meaning across the dataset. The process started with the familiarization of transcripts, which was achieved through multiple readings followed by the generation of codes by identifying meaningful pieces of text. Codes were translated into the English

language to facilitate the analysis process. After completion of coding, codes were examined for patterns and were then grouped according to similarities. Each group of similar codes was labeled with a tentative theme. The process of creating codes and themes underwent an iterative process involving co-authors (AM,MN and FF) and one qualitative expert. During the process, themes were re-examined for connections, some were merged or divided and subthemes were created. Lastly, the identified themes guided the report writing on the experiences of FCGs for hemodialysis patients with relevant quotes linked to each theme.

## Trustworthiness

Probe questions and iterative questioning were used to get detailed information while avoiding any misunderstanding. The researcher and co-authors worked together on the trail of research including verification of the data collection procedure, analysis, and interpretation process. Meetings were made virtually and physically to follow progress, deliberate on emerging issues, reflect, and plan for the next activities. Initial findings from the study were shared with some participants during subsequent visits to the clinic so that they could give their thoughts on how the findings reflected their experiences [19]. Additionally, a faculty from Muhimbili University of Health and Allied Sciences was involved in reviewing the analysis process and interpretation of data.

## Ethics approval and consent to participate

The study received ethical approval from the Institutional Review Board of Muhimbili University of Health and Allied Sciences (MUHAS) (Reference No.DA282/298/01.C/1708), with additional permission granted by Muhimbili National Hospital management. Before data collection, the researcher thoroughly explained the study's purpose, benefits of participating in the study, participants' rights and confidentiality measures to be taken. Participants were assured of their right to decline participation or skip sensitive questions without consequence. Written informed consent was obtained from each participant before conducting interviews, ensuring voluntary and transparent engagement throughout the research process.

## Results

### Participant socio-demographic characteristics

Fourteen participants were interviewed, of which nine were female and five were male. About half of the participants 7(50%) were aged between 30 and 50 years. More than one-third of participants 6(43%) had completed secondary education. Most of the participants 13(93%) had experience of caring for their patients between 1–4 years (Table 1).

During the analysis process,38 initial codes were identified, After merging similar codes, they were reduced to 10 final codes or subthemes which were further grouped into 4 main themes. The themes are 1) FCGs and health care providers' interaction, 2) The status of dialysis treatment, 3) Impact of long-term care, and 4) Coping with caring. The themes and subthemes are presented in Table 2.

### Theme 1: Family caregivers and healthcare providers' interaction

This theme provides broader experiences and perceptions of how healthcare providers interact and respond to the needs and concerns of family caregivers (FCGs) during the care process. Two sub-themes were identified: the attitude of healthcare providers and information sharing and decision-making. Generally, participants emphasized that effective

**Table 1. Participants characteristics (n=14).**

| Gender | Male | 5 |
|---|---|---|
| | Female | 9 |
| Age(Median:36 years) | Below 30 | 5 |
| | 30-49 | 7 |
| | 50-69 | 2 |
| Level of education | Primary education | 5 |
| | Secondary education | 6 |
| | College and above | 3 |
| Duration of caregiving (min: 6 months; max: 3 years; median: 17 months) | Less than a year | 1 |
| | More than a year | 13 |
| Relationship to the patient receiving hemodialysis treatment | Parent | 4 |
| | spouse | 5 |
| | Sibling | 3 |
| | Others | 2 |
| Employment status | Formal employment | 4 |
| | Self-employed | 10 |

**Table 2. Themes and subthemes related to FCGs' experience.**

| Theme | Subtheme | Codes |
|---|---|---|
| 1. Family caregivers-healthcare providers' interaction | Attitude of healthcare providers | • Lack of empathy<br>• Not listened to<br>• Supportive and helpful |
| | Information sharing and decision-making | • Inadequate shared information<br>• Delayed communication<br>• Exclusion from care decisions |
| 2. The status of dialysis treatment | Access to dialysis treatment | • High transport costs<br>• Expensive treatment<br>• Long waiting times |
| | Reliability of dialysis treatment | • Frequent machine malfunctions<br>• Staff shortages<br>• Limited dialysis machine |
| 3. Impact of long-term care | Emotional impact | • Feeling helpless<br>• Persistent frustration<br>• Loss of dignity |
| | Physical impact | • Persistent fatigue<br>• Sleep deprivation<br>• Burnout from treatment procedures |
| | Economic impact | • Financial burden of caring<br>• Reduced income-generating capacity |
| | Social impact | • Feeling abandoned<br>• Restricted life<br>• Balancing caregiving responsibilities |
| 4. Coping with caring | Religious coping | • Seeking comfort in faith<br>• Attending prayers for support |
| | Sharing experiences | • Finding comfort in talking to others<br>• Reassured through listening to similar experiences |

communication, collaboration, and empathy from healthcare providers were crucial in fostering trust and partnership with FCGs during patient care

**Attitude of healthcare providers.** There were mixed perspectives on how healthcare providers interact with FCGs. Some participants narrated that nurses had little time to talk with them when seeking information about their patients, yet they appeared busy with tasks not directly related to their work. One participant stated:

*Nurses attending to our patients should focus on their work because others are busy with other things...when you ask them something or want some clarification about the treatment, they don't pay attention to you. (P11)*

Another participant described that the way healthcare providers communicate with FCGs is situational, depending on individual character and other internal and external forces:

*It depends...you may ask a nurse a question and she may answer you with a high tone, another may answer you with a normal tone, another will smile...so it is not uniform for all. It depends on the state of their mind at that time and day. (P3)*

A similar sentiment was expressed by another participant who described that the way healthcare providers communicate depends on how they are approached:

*If you approach them in a good manner, they will respond to you well...but if you seem like challenging their profession...you will get problems and the interaction may not be smooth. (P10)*

Other participants described a good experience interacting with healthcare providers and appreciated their support and professionalism:

*The nurses and doctors have good hearts and are very supportive. I asked one doctor for his phone number to call just in case my patient had a problem, and he gave it to me without hesitation. (P1)*

In general, the subtheme describes inconsistent interactions between participants and healthcare providers, where the latter's attentiveness varied depending on their workload, personal demeanour, and the approach taken by FCGs toward engaging with them.

**Information sharing and decision-making.** Some participants emphasized the need for active sharing of relevant medical information from healthcare providers to family members. They pointed out that this aspect was lacking and asserted that timely and accurate information sharing helps FCGs make informed decisions and actively participate in supporting the patient's care:

*I need to know a lot about the procedures to follow, the plan of care, and how to manage complications associated with the disease...I should not be left speculating like this. (P4)*

Some participants thought that nurses and doctors were very busy, making meaningful interactions with patients. They recommended the allocation of more time for interactions between healthcare workers and family members:

*I think due to the high number of patients they (healthcare providers) attend, they don't have enough time for us to ask questions about our patients and procedures. We need dedicated time for meetings between doctors and us. (P7)*

The subtheme highlights participants' views that timely and transparent information sharing is essential for empowering caregivers to actively participate in care decisions. However, systemic constraints, such as time shortages, often hinder this process.

## Theme 2: The status of dialysis treatment

This theme identified two subthemes namely access to dialysis treatment and the reliability of dialysis treatment.

**Access to dialysis treatment.** Expenses related to the treatment included medical appointments, dialysis sessions, medications, laboratory tests, and transportation to the facility. High treatment costs disproportionately affected uninsured families, forcing reduced session frequency and amplifying financial strain, while insured caregivers reported relative relief.

> *It is saddening that I can only afford to bring my mother for a session once a week...sometimes I have to postpone treatment. Those with insurance can do so even thrice a day...you wish to bring your patient for treatment accordingly but you cannot afford it...the cost of treatment is very high. (P2)*

Additionally, another participant narrated:

> *The cost of treatment is not compatible with the income of an ordinary citizen...otherwise I would have brought my brother twice a week...but because of the high cost, sometimes eight days may pass...if the cost could be reduced to half...I could somehow afford. (P11)*

On the contrary, participants with insurance covering the treatment had different experiences:

> *I'm thankful that my patient has a health insurance plan that covers part of the dialysis treatment...so I'm not much stressed as I can manage other expenses. (P6)*

**Reliability of dialysis treatment.** Participants shared their experiences regarding the extent to which the treatment meets their expectations and provides optimal care for their patients:

> *You find that the room has six machines and many patients are waiting...it takes a long time until your turn comes, by then your patient's condition has worsened. (P3)*

Another participant narrated:

> *You may come here for the session and you are told there is no water (dialysate)...instead of doing the session at 10 am, you end up doing it at 4 pm... this is a problem. (P6)*

Some participants described that a shortage of staff contributes to delays:

> *I'm satisfied with the services provided; however, the problem is the shortage of staff...and sometimes nurses and doctors come late...instead of receiving treatment at 9 am, you get services at 11 am. (P12)*

Participants generally agreed that delays caused by equipment shortages, staff scarcity, and logistical disruptions negatively affected the quality of dialysis services, which in turn impacted the experiences of family caregivers

## Theme 3: Impact of long-term care

The theme focuses on understanding the emotional, physical, economic, and social impacts of caring for a loved one exhibited by FCGs.

**Emotional impact.** Caregivers reported heightened stress, mental fatigue, and a need for psychological support to navigate the emotional toll of managing chronic illness. The following quotes are expressing this:

*Caring for a kidney patient has taken a toll on my mental health due to the stress involved. Balancing the needs of an angry patient and putting on a happy face can be challenging and distressing. (P14)*

Another participant expressed:

*Because you are involved with the patient for a long time, you become mentally sick too...you may find you speak to yourself, doing abnormal things...this means we also need some kind of counselling. (P4)*

**Physical impact.** Chronic sleep deprivation and physical exhaustion were common, as FCGs prioritized patient needs over their own well-being as expressed by one participant:

*Taking care of a patient makes you unable to get more than four hours of sleep at night. In any case, you must awaken from the patient's discomfort. You'll most likely feel fatigued the next morning as a result of your lack of sleep. (P7)*

**Economic impact.** Financial sacrifices, including depleted savings and compromised livelihoods, underscored the economic difficulties faced by caregivers funding long-term treatment.This is expressed in the following excerpts:

*The process of caring for my mother has interfered with my normal life...the illness is the source of poverty. The family business has gone down because the money was just coming out to cover the treatment...but I don't blame anyone for what is happening. (P12)*

Another participant added:

*I used to think kidney failure is a disease for affluent people, but people with low life are now struck by it...Imagine you have a patient who spends a lot of money per month more than your salary...will you be able to afford treatment along with your family's needs.... you can't; it's so difficult, this is what happening to me...! (P4)*

**Social impact.** Majority of the participants said that caregiving strained familial bonds, with isolation and abandonment exacerbating the emotional and practical challenges of sustaining care. They expressed their sentiments in the following words

*Some close family members who are capable of helping are increasingly giving excuses for not being able to help our father, even being present for him. When you call them, they don't answer your phone, because they know you are going to ask them for money...so you feel you are abandoned. (P5)*

Another participant narrated that:

*The situation has left my family in shambles...when you have financial difficulties bad things start to creep into your marriage, children start behaving badly especially girls...even boys start stealing things...all because of hunger in the house. (P9)*

### Theme 4: Coping with caring

FCGs leaned on faith and peer solidarity to navigate caregiving challenges, finding solace in spiritual practices and shared experiences.

**Religious coping.** Many participants expressed a deep sense of faith, finding comfort in their belief that their circumstances were part of a divine plan. As one participant shared:

*I'm comforted by the word of God, you know everything is ordained by him, even this sickness. (P14)*

**Sharing experiences.** FCGs acknowledged that sharing experiences among fellow caregivers helped them learn from one another to deal with the hurdles of caring and therefore stay strong. One participant expressed that:

*When you have this kind of burden you wish to have someone close to you who can understand your situation, when we meet with our fellows we talk together and exchange experiences...it helps a lot to navigate through same challenges. (P9)*

### Discussion

The current study aimed to gain insight into FCGs' experience of caring for patients undergoing hemodialysis at a tertiary referral hospital in Tanzania. The FCGs described a wide range of issues resulting in four main themes: healthcare providers' interaction, the status of dialysis treatment, the impact of long-term care, and coping with care.

FCGs are partners with healthcare providers in caring for patients with chronic diseases [20]. In this relationship, healthcare providers play a crucial role in easing the burden that FCGs shoulder [21]. The findings of this study revealed that most healthcare providers did not apply empathy as a communication strategy to engage with FCGs, this could be attributed to a lack of communication skills or awareness of the important role FCGs play in patient care. The findings highlight the importance of equipping healthcare providers with competencies towards improving empathetic communication to build a trusting relationship with FCGs that can foster a good patient care experience [22–24]. This can be done by improving the organizational working environment and culture that empowers healthcare providers to demonstrate empathy [25].

FCGs identified that timely and accurate information sharing is crucial for informed decision-making and active involvement in patient care. It was noted that FCGs are not well involved in the decision-making for their patients, often they were just recipients of directives from healthcare providers of what to do for their patients. The findings of this study highlight the importance of bridging communication and collaboration gaps between healthcare providers and FCGS. Studies have shown that when healthcare providers implement the Family-Centered Care model in chronically ill patients there is improvement in communication and inclusivity in the care process therefore this should be promoted.

Caring for patients with chronic conditions affects the lives of family members, participants in this study reported disruption in their economic, physical, and psychosocial

well-being. Self-employed respondents had to sacrifice their financial activities to make time for supporting patients, and as a result, they were subjected to income hardships that compromised their ability to pay for the treatment costs of their patients. This is consistent with findings from other studies that reported financial burden as one of the major challenges for FCGs of patients with chronic diseases [26]. Literature shows that financial strain on FCGs has far-reaching consequences including creating a vicious cycle of dependency and being a root cause for other health-related problems in FCGs including depression [11,27–30].

Participants reported physical exhaustion attributed to fatigue and sleep disruptions, due to frequent visits for hemodialysis sessions several times per week. The findings of this study are consistent with previous studies indicating fatigue to be a common physical problem affecting the quality of life among FCGs [31–33]. Moreover, our study revealed that FCGs experienced heightened levels of anxiety and distress related to meeting the treatment demands of their patients. The finding of our study concurs with past research showing that depression, anxiety, and feeling burdened were common among FCGs of patients with chronic illnesses [34–37]. The findings highlight the importance of having strategies in place to mitigate these consequences. Such strategies may include educational, supportive, and counseling interventions to build resiliency and capacity to handle the situation. In addition, FCGs should be encouraged to practice their religious freedom and share experiences among themselves as a means of coping with caring experiences [9,26,38,39].

The findings of the study show most FCGs struggle to bear the cost of dialysis treatment considering that the majority of them had no insurance schemes and therefore had to use cash. In this context, some prescribed medications and therapy sessions are skipped affecting the effectiveness of treatment. The findings of our study are similar to previous studies reporting that high dialysis treatment costs lead to a financial burden on FCGs resulting in poor adherence to dialysis treatment among patients [27,40–42]. Our findings show the need to implement health policies and strategies that can lower treatment costs to benefit the majority of treatment users who happen to be low-income earners. This must go in hand with increasing the number of dialysis centers and dialysis machines, which are currently scarce, compared to the number of patients. Improvement in the availability and quality of dialysis treatment is anticipated to increase access and reduce anxiety among FCGs [43].

## Practical implications of research findings

The findings of this study have important implications for improving the support and well-being of FCGs of patients undergoing hemodialysis. The study highlights the critical role of healthcare providers in fostering a collaborative and empathetic relationship with FCGs to enhance patient care. A lack of empathetic communication and limited involvement of FCGs in decision-making suggest targeted training programs to equip healthcare providers with effective communication skills and a family-centred care approach. Additionally, the financial, physical, and emotional burdens faced by FCGs emphasize the necessity for policy interventions, such as subsidized treatment costs and increased access to dialysis facilities. The findings also underscore the importance of psychosocial support strategies, including counselling services and peer support to address the anxiety and financial strain reported by FCGs. By addressing these challenges, healthcare systems can improve the caregiving experience, enhance patient outcomes, and ensure a supportive healthcare environment for both patients and their caregivers.

## Strengths and limitations of the study

This may be the first study exploring the experiences of FCGs of patients undergoing hemodialysis in a Tanzanian setting, offering an in-depth insight into the situation and

therefore filling some knowledge gaps regarding the provision of dialysis services within the healthcare system. It is crucial to recognize that the interviews were conducted in one dialysis unit in Tanzania and therefore experiences of FCGs regarding the services and interactions with healthcare providers may not be the same limiting the transferability of the findings. Considering that the nature of the research required the FCGs to recall their experiences, this could have subjected data to recall bias. Nevertheless, principles of ensuring the trustworthiness of the study were rigorously followed. Moreover, as the interviews were conducted in Kiswahili and later translated into English at the coding level, there is a risk of misinterpretation or loss of meaning during translation.

## Conclusion

Family caregivers of hemodialysis patients experience some challenges interacting and communicating with healthcare providers. There is a need to improve mutual relationships and involvement in the care process to optimize patient outcomes. Additionally, the caring process subjects FCGS to a burden encompassing economic, physical, and psychosocial. To effectively cope with the burden, tailored supportive interventions embracing a family-centered model need to be implemented or enhanced.

## Supporting information

**S1 Text. A COREQ checklist.**
(DOCX)

**S2 Text. A guide for in-depth interview.**
(DOCX)

**S3 Text. Excerpts from transcripts.**
(DOCX)

## Acknowledgements

The authors would like to thank all study participants and the Muhimbili National Hospital management for facilitating data collection.

## Author contributions

**Conceptualization:** Ally Halfan Magenge, Menti Lastone Ndile.

**Formal analysis:** Ally Halfan Magenge, Menti Lastone Ndile.

**Methodology:** Ally Halfan Magenge, Menti Lastone Ndile.

**Writing – original draft:** Ally Halfan Magenge, Menti Lastone Ndile.

**Writing – review & editing:** Ally Halfan Magenge, Menti Lastone Ndile, Francis Furia.

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
