## [Decision Letter · Decision Letter 0]

15 Jan 2025

PONE-D-24-43236Family caregivers' experience of caring for patients undergoing hemodialysis at Muhimbili National Hospital, Dar es Salaam, TanzaniaPLOS ONE

Dear Dr. Ndile,

Thank you for submitting your manuscript to PLOS ONE. After careful consideration, we feel that it has merit but does not fully meet PLOS ONE’s publication criteria as it currently stands. Therefore, we invite you to submit a revised version of the manuscript that addresses the points raised during the review process.

We look forward to receiving your revised manuscript.

Kind regards,

Deogratias Munube

Academic Editor

PLOS ONE

Journal Requirements:

Please confirm at this time whether or not your submission contains all raw data required to replicate the results of your study. Authors must share the “minimal data set” for their submission. PLOS defines the minimal data set to consist of the data required to replicate all study findings reported in the article, as well as related metadata and methods (https://journals.plos.org/plosone/s/data-availability#loc-minimal-data-set-definition ).

If your submission does not contain these data, please either upload them as Supporting Information files or deposit them to a stable, public repository and provide us with the relevant URLs, DOIs, or accession numbers. For a list of recommended repositories, please see https://journals.plos.org/plosone/s/recommended-repositories .

3. Please include captions for your Supporting Information files at the end of your manuscript, and update any in-text citations to match accordingly. Please see our Supporting Information guidelines for more information: http://journals.plos.org/plosone/s/supporting-information .

Reviewers' comments:

Reviewer's Responses to Questions

**Comments to the Author**

1. Is the manuscript technically sound, and do the data support the conclusions?

Reviewer #1: Yes

Reviewer #2: Partly

2. Has the statistical analysis been performed appropriately and rigorously? 

Reviewer #1: N/A

Reviewer #2: Yes

3. Have the authors made all data underlying the findings in their manuscript fully available?

Reviewer #1: No

Reviewer #2: No

4. Is the manuscript presented in an intelligible fashion and written in standard English?

Reviewer #1: Yes

Reviewer #2: Yes

5. Review Comments to the Author

Reviewer #1: Thank you for the opportunity to review this research. This article investigates the experience of family caregivers of patients undergoing hemodialysis in a qualitative phenomenological study.

I would like to draw the authors’ attention to the following comments:

• The title should mention the type of study.

Abstract

• An in-depth interview guide was used for data collection.

• It seems the word “guide” should be removed here.

• In the results section of the abstract, it should be specified how many individuals were interviewed, how many codes were extracted, and additional information should be provided to give the reader a clearer understanding.

• More keywords can be added, such as "caregiver burden," "care," etc.

Problem Statement

• The knowledge gap has not been adequately addressed in the problem statement. Based on the current literature, what do we know and what do we not know?

• At the end of the problem statement, it should be explained in a paragraph why a qualitative research method is suitable for this study.

Methods

• It should be explicitly mentioned where the interviews were conducted.

• How did you ensure data saturation was achieved?

• Were any interviews repeated?

• During the interviews, was there anyone else present besides the interviewer and the participant?

• Did anyone withdraw from participating in the interview?

• How was coding agreement reached?

• What was your coding unit? In addition to interview content, were other data sources, such as field notes, used?

• A larger sample of interview questions should be provided.

• Attach the interview guide.

• This section primarily addresses research ethics rather than the rigor of the study:

"To ensure the trustworthiness and credibility of research findings, the researcher had time to establish rapport with the participants and explain the purpose of the study; this established a trustful relationship between the researcher and individual participants. Participants were also informed that they were free to decline to participate, this ensured honesty in the voluntarily provided information."

• It seems the rigor (trustworthiness) section needs further elaboration. Was peer-checking conducted?

Ethical Considerations

• Was permission obtained to record the interviews?

• Were participants allowed to refrain from answering any question they were uncomfortable with? Were they assured of confidentiality? These points should be mentioned in the ethical considerations.

Results

• The number of initial codes obtained should be specified, and then, after merging similar ones, the number of final codes should be stated.

• Each theme should have at least three sub-themes.

• The level of abstraction of the codes is very high.

Discussion/Conclusion

• It would be better to discuss the practical implications of the research findings at the end of the discussion or in the conclusion section.

Reviewer #2: Comments

Very interesting study with relevant results pertinent to policy and action in low- and middle-income countries.

-Background: Need to improve this section by adding some global data and statistics from the global burden of diseases.

Some areas need focused attention and further development – p. 5 The main question was Tell me about your life when you started caring for your loved one in need of hemodialysis treatment. Unclear how one question generated the themes/subthemes- it will be helpful to attach data collection instrument – i.e. Interview Guide.

- Clarity is needed on recruitment and sampling procedures; did the researchers just approach the FCG and talked to them about the study? At that point in time were the participants recruited into the study? Were there any benefits for the study participants?

- Was the sampling just purposive, how about convenience sampling?

- What was the study’s inclusion and exclusion criteria?

- P. 5 - All Interviews were conducted in Kiswahili language which is the national language of the participants.-how did the translation to English affect data quality, what measures were put in place to ensure methodological rigor. How does this become study limitation noting that words in the local language and its translation into English does not always have its exact meaning in the English language.

- Need to number the themes and subthemes – the subthemes are quite not exhaustive – there is little information on system and structural support

- P.10 – main text and participant quotes need to be differentiated well – they appear lumped up. need to present quotes with due indents.

- The statements in between quotes need to be well presented, as best practice, these statements should be provided in between quotes and especially before transitioning to a theme or subthemes. Almost all the quotes ended without transition statements and new themes/subthemes were introduced.

- In relation to Employment status, was there any participant that was unemployed?

- Based on the study findings, the recommendations need to be quite clear – what are the implications for policy, practice and planning?

6. PLOS authors have the option to publish the peer review history of their article (what does this mean? ). If published, this will include your full peer review and any attached files.

**Do you want your identity to be public for this peer review?** For information about this choice, including consent withdrawal, please see our Privacy Policy .

Reviewer #1: No

Reviewer #2: **Yes: ** Mary Ani-Amponsah

---

## [Author Response · Author response to Decision Letter 1]

25 Feb 2025

Comments to reviewers and the editor have been very constructive and all incorporated in the revised version. The detail is attached in response to the reviewer's comments file

---

## [Editor Report · Decision Letter 1]

12 Mar 2025

Family Caregivers’ Experience of Caring for Patients Undergoing Hemodialysis: A Qualitative Study at Muhimbili National Hospital in Dar es Salaam, Tanzania

PONE-D-24-43236R1

Dear Dr. Menti Lastone Ndile

We’re pleased to inform you that your manuscript has been judged scientifically suitable for publication and will be formally accepted for publication once it meets all outstanding technical requirements.

Kind regards,

Deogratias Munube

Academic Editor

PLOS ONE

Additional Editor Comments (optional):

Dear Author,

Thank you for addressing the concerns raised by the reviewers.
---

## [Editor Report · Acceptance letter]

PONE-D-24-43236R1

PLOS ONE

Dear Dr. Ndile,

I'm pleased to inform you that your manuscript has been deemed suitable for publication in PLOS ONE. Congratulations! Your manuscript is now being handed over to our production team.

Kind regards,

on behalf of

Dr. Deogratias Munube

Academic Editor

PLOS ONE